# Role of YAP in early ectodermal specification and a Huntington's Disease model of human neurulation

Francesco M Piccolo[1†], Nathaniel R Kastan[2,3†], Tomomi Haremaki[1§], Qingyun Tian[1], Tiago L Laundos[1,4,5,6], Riccardo De Santis[1], Andrew J Beaudoin[1], Thomas S Carroll[7], Ji-Dung Luo[7], Ksenia Gnedeva[2,3#], Fred Etoc[1], AJ Hudspeth[2,3‡], Ali H Brivanlou[1*‡]

[1]Laboratory of of Stem Cell Biology and Molecular Embryology, The Rockefeller University, New York, United States; [2]Howard Hughes Medical Institute, The Rockefeller University, New York, United States; [3]Laboratory of Sensory Neuroscience, The Rockefeller University, New York, United States; [4]ICBAS - Instituto de Ciências Biomédicas Abel Salazar, Universidade do Porto, Porto, Portugal; [5]i3S - Instituto de Investigação e Inovação em Saúde, Universidade do Porto, Porto, Portugal; [6]INEB - Instituto de Engenharia Biomédica, Universidade do Porto, Porto, Portugal; [7]Bioinformatics Resource Center, The Rockefeller University, New York, United States

**\*For correspondence:** brvnlou@mail.rockefeller.edu

†These authors contributed equally to this work
‡These authors also contributed equally to this work

**Present address:** §RUMI Scientific, New York, United States; #Tina and Rick Caruso Department of Otolaryngology—Head and Neck Surgery, University of Southern California, Los Angeles, United States

**Abstract** The Hippo pathway, a highly conserved signaling cascade that functions as an integrator of molecular signals and biophysical states, ultimately impinges upon the transcription coactivator Yes-associated protein 1 (YAP). Hippo-YAP signaling has been shown to play key roles both at the early embryonic stages of implantation and gastrulation, and later during neurogenesis. To explore YAP's potential role in neurulation, we used self-organizing neuruloids grown from human embryonic stem cells on micropatterned substrates. We identified YAP activation as a key lineage determinant, first between neuronal ectoderm and nonneuronal ectoderm, and later between epidermis and neural crest, indicating that YAP activity can enhance the effect of BMP4 stimulation and therefore affect ectodermal specification at this developmental stage. Because aberrant Hippo-YAP signaling has been implicated in the pathology of Huntington's Disease (HD), we used isogenic mutant neuruloids to explore the relationship between signaling and the disease. We found that HD neuruloids demonstrate ectopic activation of gene targets of YAP and that pharmacological reduction of YAP's transcriptional activity can partially rescue the HD phenotype.

## Editor's evaluation

This manuscript harnesses an organoid model of human neurulation to unravel the role of the Hippo signalling pathway in the specification of the three key ectodermal cell types. The authors then investigate how these mechanisms are dysregulated in an organoid model of Huntington's Disease. This work will be of broad interest to readers interested in the process of neurulation and how dysregulation of developmental mechanisms may lead to disease conditions in adulthood.

## Introduction

Although the study of embryogenesis benefits from a broad array of model organisms and the conservation of mechanisms across species, elucidation of the particularities of human development remains challenging. The use of human pluripotent stem cells for the creation of synthetic human organoids

has greatly enhanced our ability to investigate human development (*Kim et al., 2020*). For example, the process of neurulation can be mimicked in vitro through micropattern-based neuruloids (*Haremaki et al., 2019*). This technique, which allows the generation of hundreds of virtually identical organotypic cultures, offers a powerful opportunity to investigate the molecular and biophysical principles of human neurulation and the associated diseases.

The Hippo pathway is an ancient and highly conserved signaling cascade that operates in numerous cell types and a variety of organisms. In contrast to developmental pathways that require specific ligand-receptor interactions, the Hippo pathway functions as an integrator of molecular signals and biophysical states, including cell polarity, adhesion, GPCR signaling, and mechanical forces sensed through the cytoskeleton. With key roles in development, homeostasis, and regeneration, the pathway interacts with other developmental and regulatory signals including Wnt, Notch, and TGFβ. Despite the apparent complexity, the core cascade is relatively simple, including the parallel mammalian STE20-like kinases 1 and 2 (MST1/2) and mitogen-activated protein kinase kinase kinase kinase (MAP4K). The activity of these enzymes leads to phosphorylation of large tumor suppressor kinases 1 and 2 (LATS1/2), which in turn phosphorylate and limit the nuclear localization of Yes-associated protein 1 (YAP) and the homologous WW domain-containing transcription regulator 1 (TAZ). When they translocate to the nucleus, YAP and TAZ cooperate with TEA-domain transcription factors (TEAD1/2/3/4) to activate the expression of a variety of genes associated with proliferation, cell survival, de-differentiation, and cellular morphogenesis (*Davis and Tapon, 2019*; *Hansen et al., 2015*; *Juan and Hong, 2016*; *Moya and Halder, 2019*).

Hippo-YAP signaling is essential at various stages of embryonic development, including the zygote, early blastomeres, the inner cell mass (ICM), trophectoderm (TE), and the emergence of individual organs (*Wu and Guan, 2021*). Because Yap$^{-/-}$ mice die at E8.5 owing to the failure of endothelium formation in the yolk sac, and Yap$^{-/-}$, Taz$^{-/-}$ embryos fail prior to the morula stage, investigating the potential role of Yap in neurulation in vivo requires complex transgenic strategies (*Morin-Kensicki et al., 2006*; *Nishioka et al., 2009*). The use of neuruloids derived from human embryonic stem cells (hESCs) therefore offers a convenient, highly reproducible, and quantitative alternative means of investigating the role of Hippo signaling in neurulation.

Here, we identify YAP activation as a key lineage determinant, first between neuronal ectoderm (NE) and nonneuronal ectoderm (NNE), and later between epidermis and neural crest (NC), in human embryo models. Exploiting the reproducible phenotype of Huntington's Disease (HD) mutations in neuruloids, we additionally find that HD neuruloids demonstrate stronger YAP's downstream transcriptional signature and that the pharmacological reduction of YAP's transcriptional activity partially reverses the HD phenotype.

## Results

### Differential YAP activity in neural and nonneural ectoderm

In order to study the role of the Hippo pathway during human neurulation in vitro, we applied a neuruloid assay (*Haremaki et al., 2019*). Pluripotent hESCs were grown on circular, 500-µm diameter circular substrates and converted to ectodermal fates by dual SMAD inhibition for 3 days (SB431542 and LDN193189, SB+LDN, and neural induction) followed by removal of LDN and BMP4 stimulation (SB+BMP4 and neurulation). As previously reported, stimulation until the fourth day after plating, which we designate hereafter as D4, led to the induction of NNE at the edge and NE in the center (*Haremaki et al., 2019*; *Figure 1A*).

To assess Hippo-YAP signaling during these early stages of neuruloid formation, we genetically engineered the RUES2 hESC line (National Institutes of Health #0013) to endogenously and bialleli-cally tag the C-terminus of YAP protein with green fluorescent protein (GFP; *Figure 1—figure supplement 1* and *Figure 1—figure supplement 2*; *Franklin et al., 2020*). During the pluripotency state at the outset of culture, at D0, YAP was highly expressed in the nuclei, and progressively enriched in the cytoplasm upon neural induction. YAP expression was downregulated by D3 (*Figure 1B and C*).

Live imaging 24 hr after BMP4 stimulation (D4), revealed upregulation of YAP expression at the edge of the colony, in the region where the NNE lineage would emerge. This upregulation at the periphery was BMP4-dependent. At this stage, the upregulation of YAP at the periphery depended on BMP4 stimulation, which might directly regulate YAP expression, for SMAD1, the transcription

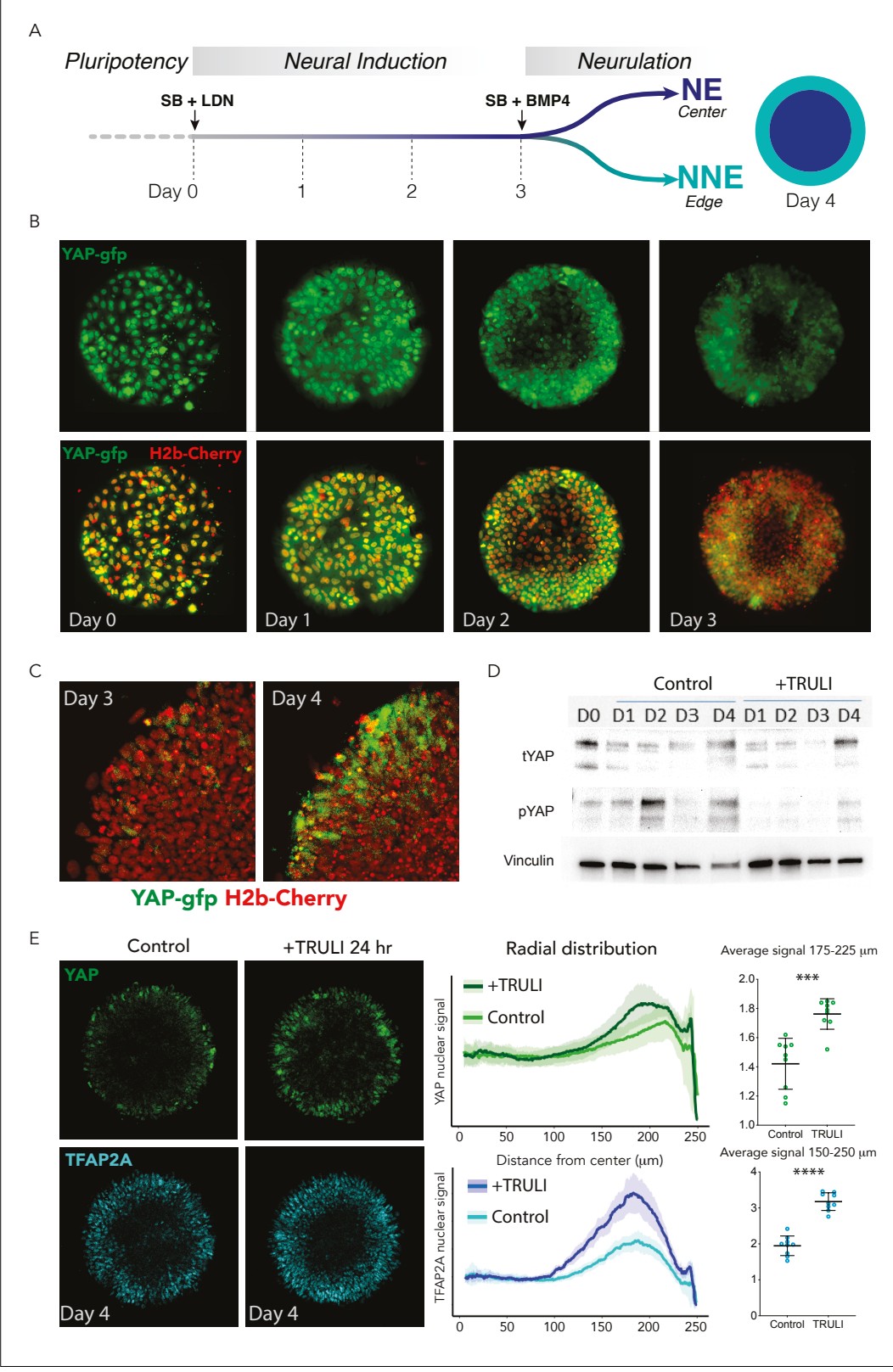

**Figure 1.** Dynamics of YAP expression and localization in early neuruloids. (**A**) During the first 4 days of the differentiation protocol, pluripotent hESCs seeded on a micropattern undergo 3 days of neural induction (SB +LDN) followed by BMP4 induction of neurulation (SB +BMP4). (**B**) Fluorescent images of living YAP-GFP and H2b-Cherry hESC colonies during neural induction show the progressive shift of YAP from nuclei into cytoplasm

*Figure 1 continued on next page*

*Figure 1 continued*

and a reduction in expression level. (**C**) Fluorescent images portray one quadrant of a colony acquired before (D3) and 24 hr after (D4) BMP4 administration. (**D**) Immunoblots from D0 to D4 of the neuruloid protocol illustrate the decrease in YAP phosphorylated at residue S127 (pYAP) upon treatment with 10 μM TRULI. The amount of total YAP protein (tYAP) initially declines but recovers by D4. Vinculin serves as a loading control. (**E**) Left: Immunolabeling of D4 neuruloids demonstrates the increased concentration of YAP and TFAP2A, a marker of NNE, at the edges of colonies treated for 24 hr with 10 μM TRULI. Right: Radial distribution of YAP and TFAP2A nuclear signals from several micropattern colonies quantitate the result. The average values at the edge of each colony are also shown. ****, p<0.0001; ***, p<0.001 in unpaired t-tests comparing untreated (control) and TRULI-treated samples. Nine colonies were analyzed for each condition. hESC, human embryonic stem cell.

The online version of this article includes the following video, source data, and figure supplement(s) for figure 1:

**Source data 1.** Raw images for immunoblot shown in *Figure 1D*.

**Figure supplement 1.** Screening of WT and HD YAP-GFP reporter lines.

**Figure supplement 2.** Validation of WT and HD YAP-GFP reporter lines.

**Figure supplement 3.** BMP4-dependent upregulation of YAP in D4 neuruloids.

**Figure supplement 4.** Quantification of immunoblots of total YAP.

**Figure supplement 5.** Magnified images of D4 neuruloids labeled for YAP and DAPI.

**Figure 1—video 1.** Live YAP dynamics in D3 WT neuruloid upon BMP4 stimulation.
https://elifesciences.org/articles/73075/figures#fig1video1

**Figure 1—video 2.** Live YAP dynamics in D3 WT neuruloid upon BMP4 stimulation (detail, YAP-GFP and H2b-mCherry).
https://elifesciences.org/articles/73075/figures#fig1video2

**Figure 1—video 3.** Live YAP dynamics in D3 WT neuruloid upon BMP4 stimulation (detail, YAP-GFP).
https://elifesciences.org/articles/73075/figures#fig1video3

**Figure 1—video 4.** Live YAP dynamics in D3 WT neuruloid upon BMP4 stimulation and activation by TRULI.
https://elifesciences.org/articles/73075/figures#fig1video4

**Figure 1—video 5.** Live YAP dynamics in D3 WT neuruloid upon BMP4 stimulation and activation by TRULI (Detail, YAP-GFP and H2b-mCherry).
https://elifesciences.org/articles/73075/figures#fig1video5

**Figure 1—video 6.** Live YAP dynamics in D3 WT Neuruloid upon BMP4 stimulation and activation by TRULI (Detail, YAP-GFP).
https://elifesciences.org/articles/73075/figures#fig1video6

factor downstream of BMP4 signaling, appeared to be enriched in its promoter and regulatory regions (*Figure 1—figure supplement 3A and B*). Near the border of the colony, YAP could be observed to undergo sporadic nuclear enrichment (*Figure 1C*; *Figure 1—video 1*; *Figure 1—video 2*, *Figure 1—video 3*) suggesting dynamic regulation of the protein's localization in the developing NNE lineage.

## YAP activation with TRULI augments differentiation of nonneural ectoderm

Immunoblot analyses of D4 neuruloids confirmed a rise in YAP levels and revealed phosphorylation of YAP at residue S127, consistent with regulation through the Hippo pathway (*Figure 1D*). Concomitant application of the Lats-kinase inhibitor TRULI (*Kastan et al., 2021*) suppressed YAP phosphorylation without altering the dynamics of the protein's concentration during this initial phase of neuruloid self-organization (*Figure 1D*). The immunoblots also showed that at the pluripotent stage (D0) YAP was expressed as two protein isoforms (*Figure 1D*; *Figure 1—figure supplement 1*), both downregulated during neural induction (D1–D3; *Figure 1D*; *Figure 1—figure supplement 4A*). The upregulation upon BMP4 induction primarily involved the larger, upper band.

Consistent with the loss of S127 phosphorylation, administration of TRULI in conjunction with BMP4 increased the level of YAP protein and induced its nuclear localization in cells at the edges of neuruloid colonies (*Figure 1E*; *Figure 1—figure supplement 5*; *Figure 1—video 4*, *Figure 1—video 5* and *Figure 1—video 6*). Finally, TRULI-treated neuruloids demonstrated more robust TFAP2A expression, suggesting enhanced NNE induction (*Figure 1E*). These results together suggest that: YAP expression

is upregulated in response to BMP4 at the periphery of the colony and is subsequently regulated in part by the Hippo pathway, contributing directly to induction of the NNE lineage.

## YAP activity contributes to differentiation of neural crest and epidermis

After the initial separation of the NE and NNE lineages at D4, maturation to a complete neuruloid occurs by D7 (*Haremaki et al., 2019*). In the course of this process, the NE organizes into a compact,

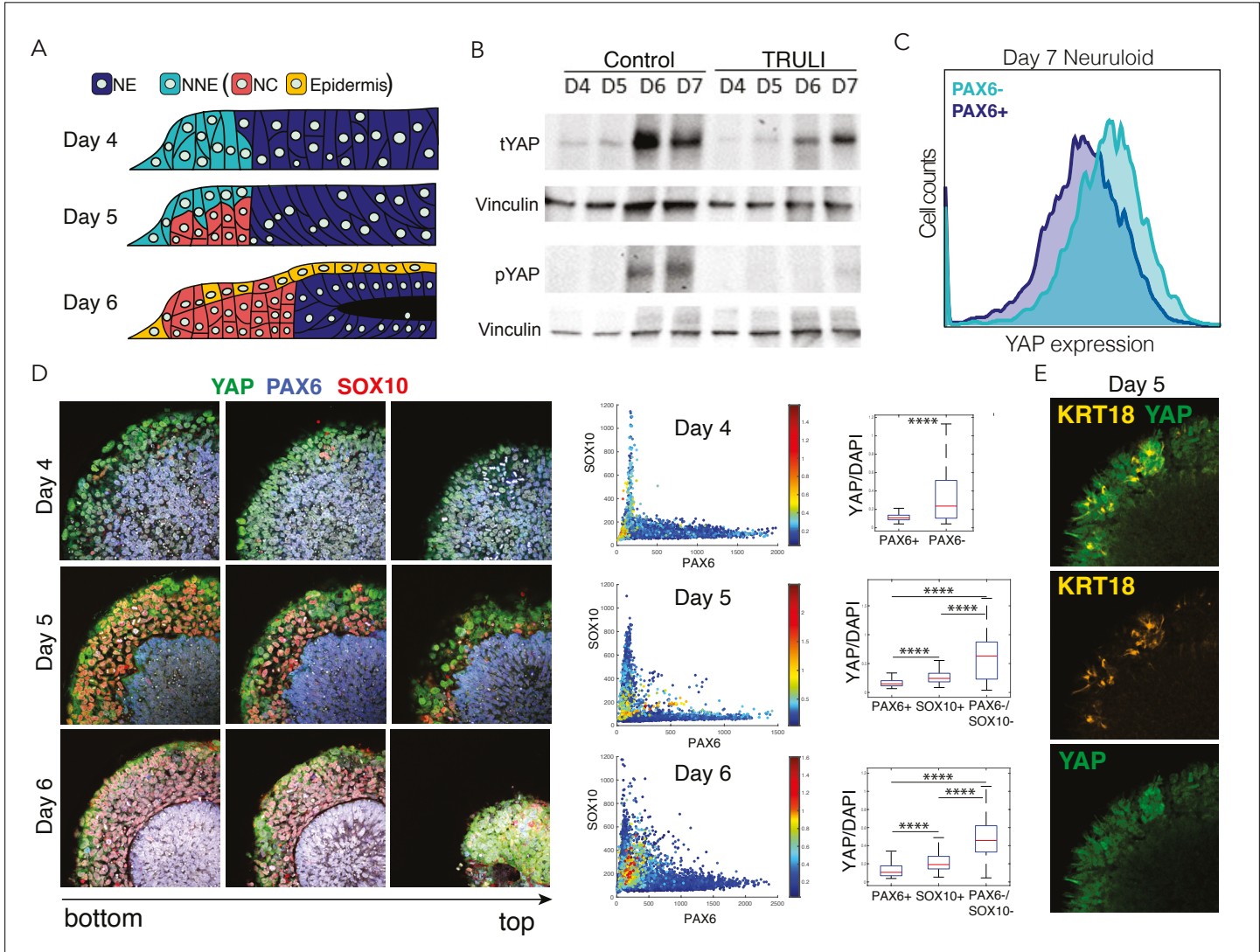

**Figure 2.** Dynamics of YAP expression and localization in late neuruloids. (**A**) In a schematic side view of the final phase of neuruloid formation, neural ectoderm (NE, dark blue) at the center is surrounded by nonneural ectoderm (NNE, cyan), which subsequently differentiates into neural crest (NC, red) and epidermis (yellow). (**B**) Immunoblot analysis of neuruloids at D4–D7 demonstrates the suppression of YAP phosphorylated on residue S127 (pYAP) by 10 µM TRULI. The concentration of total YAP protein (tYAP) increases under both conditions. Vinculin provides a loading control. (**C**) Analysis by fluorescence-activated cell sorting of D7 neuruloids shows less expression of total YAP protein in NE (PAX6+) than in NNE (PAX6−). The difference in YAP expression level between the two lineages was analyzed by both two-way ANOVA and nonparametric t-tests, consistently showing p<0.000001. (**D**) Immunohistochemistry during the late phase of neuruloid formation, D4–D6, shows the progressive decline in nuclear YAP in NE and NC lineages. For four neuruloid colonies at each time point, the nuclear signals of YAP, PAX6, and SOX10 were determined in individual cells. Scatterplots and box plots of normalized YAP signals quantify the effect. ****, p<0.0001 in unpaired nonparametric t-tests comparing the level of normalized nuclear YAP signal between the different conditions. (**E**) Immunofluorescence images of one quadrant of a D5 neuruloid demonstrate the presence of KRT18, a label for epidermis, in cells enriched for nuclear YAP.

The online version of this article includes the following source data and figure supplement(s) for figure 2:

**Source data 1.** Raw images for immunoblot shown in *Figure 2B*.

**Figure supplement 1.** Dynamics of YAP expression and localization in late neuruloids (magnified).

closed structure at the center of the colony. The NNE differentiates further, forming two principal derivatives: NC, which is found in a radially symmetric arrangement at the periphery of the neuroloid, and epidermal cells (E), which grow atop the culture (*Figure 2A*). The overall structure models the early stages of human neurulation (*Haremaki et al., 2019*). On D6, YAP levels rose sharply with a concomitant increase of phospho-S127 YAP (*Figure 2B*; *Figure 1—figure supplement 4B*). The effect predominantly involved the larger isoform, which continued to be upregulated following BMP4 induction. Fluorescence-activated cell sorting on D7 demonstrated that enrichment of YAP in NNE (PAX6−) relative to NE (PAX6+) persisted through maturation of the neuroloid (*Figure 2C*).

We assessed the expression and localization of YAP within the major ectodermal lineages at different times during neuroloid formation (*Figure 2D and E*; *Figure 2—figure supplement 1*). This confirmed the exclusion of YAP from the nuclei both of NE (PAX6+) cells and of the emerging neural-crest cells (SOX10+; *Figure 2D*). YAP was most notably present in the nuclei of cells expressing the early epidermal marker keratin 18 (KRT18; *Figure 2E*).

Given that YAP activity is enriched first in undifferentiated NNE and subsequently in epidermis, we inquired whether YAP signaling stimulates their differentiation. Taking advantage of a protocol for the direct differentiation of NC (*Tchieu et al., 2017*), we produced SOX10+ colonies with only sparse KRT18+ cells (*Figure 3—figure supplement 1A*). The administration of TRULI, which enhanced YAP activity (*Figure 1E* and *Figure 1—figure supplement 2*), resulted in the complete loss of NC-like SOX10+ colonies and a significant increase in KRT18 expression (*Figure 3—figure supplement 1A*). Consistent with these observations, in a converse experiment TRULI treatment boosted KRT18 expression in an epidermal-differentiation protocol (*Figure 3—figure supplement 1B*; *Tchieu et al., 2017*). Furthermore, TRULI-treated neuroloids displayed increased differentiation toward epidermis and a reduction in SOX10+ cells (*Figure 3*). Surprisingly, TRULI treatment also precluded the closure of the central PAX6+ domain, the simulacrum of the neural tube, despite its being devoid of detectable YAP (*Figure 3*).

These experiments highlight the role of YAP in an in vitro model of human neurulation and suggest that the protein is dynamically regulated through canonical Hippo signaling during the specification of ectodermal lineages. Following an initial downregulation in ectodermal cells, the expression of YAP protein is induced by BMP4 stimulation and its activity supports the induction of the NNE lineage. YAP subsequently contributes directly to the differentiation of NNE by repressing the development of the NC lineage and promoting the emergence of epidermal cells. Excessive YAP activity therefore affects the balance in specification of NC and epidermis and hinders the ability of the central NE domain to fully compact.

## Developing HD neuroloids display increased YAP activity

HD is caused by an expansion of CAG repeats in the Huntington locus (*HTT*). The development of HD can be modeled in RUES2 isogenic lines that are engineered to introduce CAG expansion, generating a unique phenotypic signature in HD-neuroloids (*Haremaki et al., 2019*). Using this approach, we characterized the phenotype of HD-neuroloids made from RUES2 bearing 56 CAG repeats. Interestingly, HD-neuroloids generate the same signature as TRULI-treated neuroloids: both fail to display closure of the central PAX6+ area (*Figure 3*). Intrigued by the similarities between the two phenotypes, and in view of the potential roles of YAP in human ectodermal differentiation, we next investigated whether the HD phenotype in neuroloids stems from hyperactivity of YAP.

To directly compare the behavior of YAP in WT and HD neuroloids, we genetically engineered an HD line to express endogenously tagged YAP-GFP (*Figure 1—figure supplement 1* and *Figure 1—figure supplement 2*). The behavior of YAP in WT and HD neuroloids was indistinguishable during the first 3 days of neural induction, when YAP was excluded from nuclei and progressively downregulated (*Figure 4—figure supplement 1*). Both WT and HD lines likewise upregulated YAP expression at the edges of colonies upon BMP4 stimulation. However, at D4 YAP appeared to be more enriched in nuclei at the most peripheral region of HD-neuroloids compared to WT controls (*Figure 4A*; *Figure 4—video 1*, *Figure 4—video 2*, *Figure 4—video 3*). A similar result was obtained by immunofluorescence microscopy, which confirmed elevated YAP nuclear signals in HD colonies (*Figure 4B*; *Figure 1—figure supplement 5*). Consistent with these observations, immunoblotting analysis of D4 HD neuroloids revealed a reduction in the phosphorylated form of YAP in comparison to WT controls (*Figure 4C*). These observations suggest increased YAP activity in early HD neuroloids.

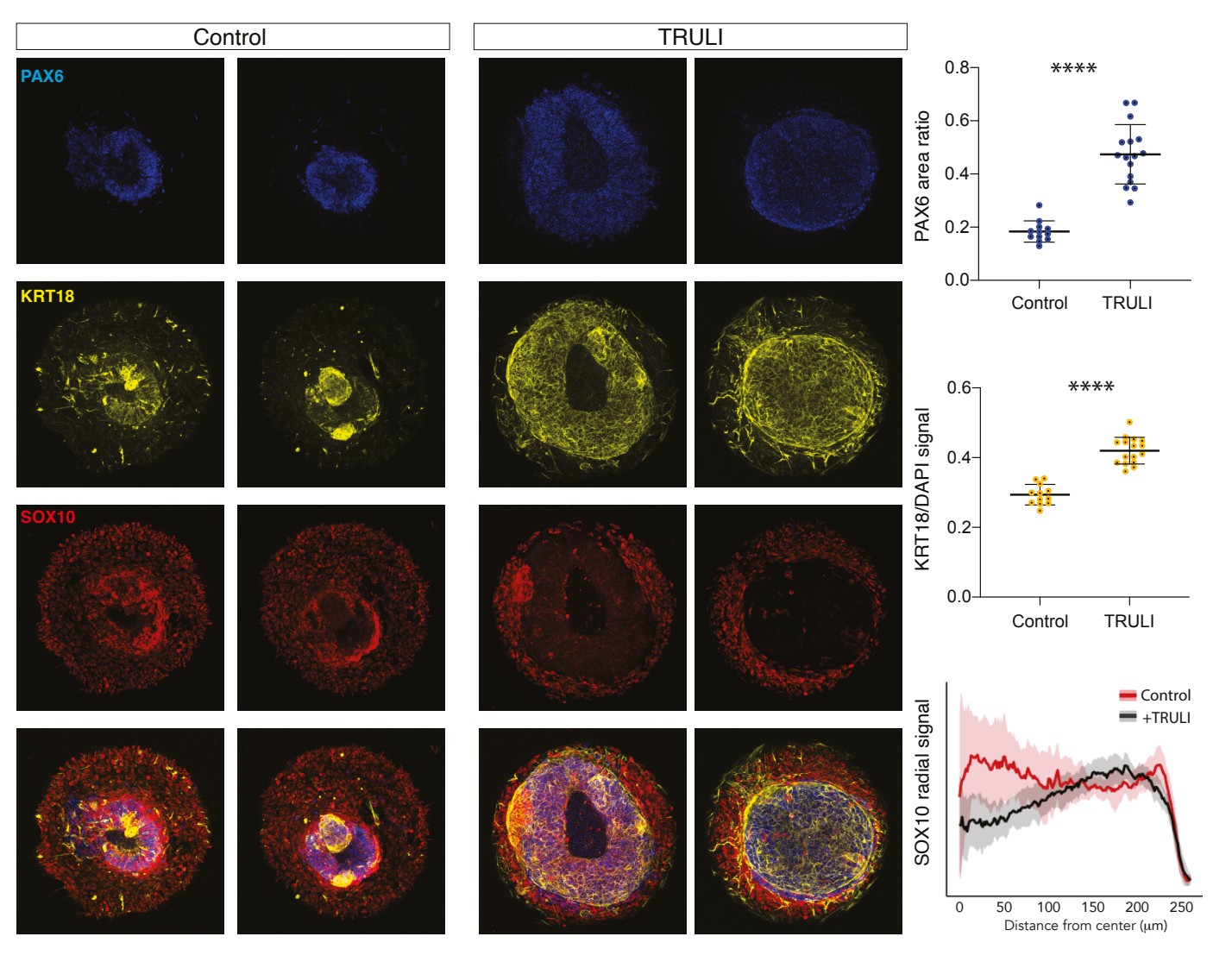

**Figure 3.** Effect of sustained YAP activation on neural crest and neural epithelium. Immunofluorescence images of D7 WT neuroloids demonstrate that treatment with 10 µM TRULI enhances the expression of neural ectoderm marked by PAX6 and of epidermis marked by KRT18 at the expense of NC labeled by SOX10. The plots quantify the fractions of the areas of several colonies marked by PAX6 and KRT18 (as normalized to DAPI), as well as the radial distribution of SOX10. ****, p<0.0001 in an unpaired t-test comparing untreated (control, 12 colonies analyzed) and TRULI-treated samples (TRULI, 16 colonies analyzed).

The online version of this article includes the following figure supplement(s) for figure 3:

**Figure supplement 1.** Effect of sustained YAP activation on differentiation of neural crest (NC) and epidermis.

As observed after TRULI treatment of WT RUES2 cells (*Figure 1E*), D4 HD neuroloids displayed more TFAP2A at the edges of colonies (*Figure 4D*; *Figure 4—figure supplement 2*). This observation implies that increased YAP activity in early HD neuroloids has early and direct consequences for fate acquisition in the developing NNE. Consistent with elevated YAP activity, by D7, both HD neuroloids and TRULI-treated WT neuroloids displayed a remarkably expanded epidermal lineage and a reduction in the neural-crest population, as well as a failure in NE compaction (*Figure 4E*).

These results indicate that neuroloids obtained from hESCs carrying an HD mutation display elevated YAP activity. Consistent with our observation that YAP activity regulates NNE and epidermal formation, at D4, the HD neuroloids demonstrate more robust induction of NNE, which results by D7 in significant expansion of the epidermis at the expense of NC and in incomplete closure of the central NE domain.

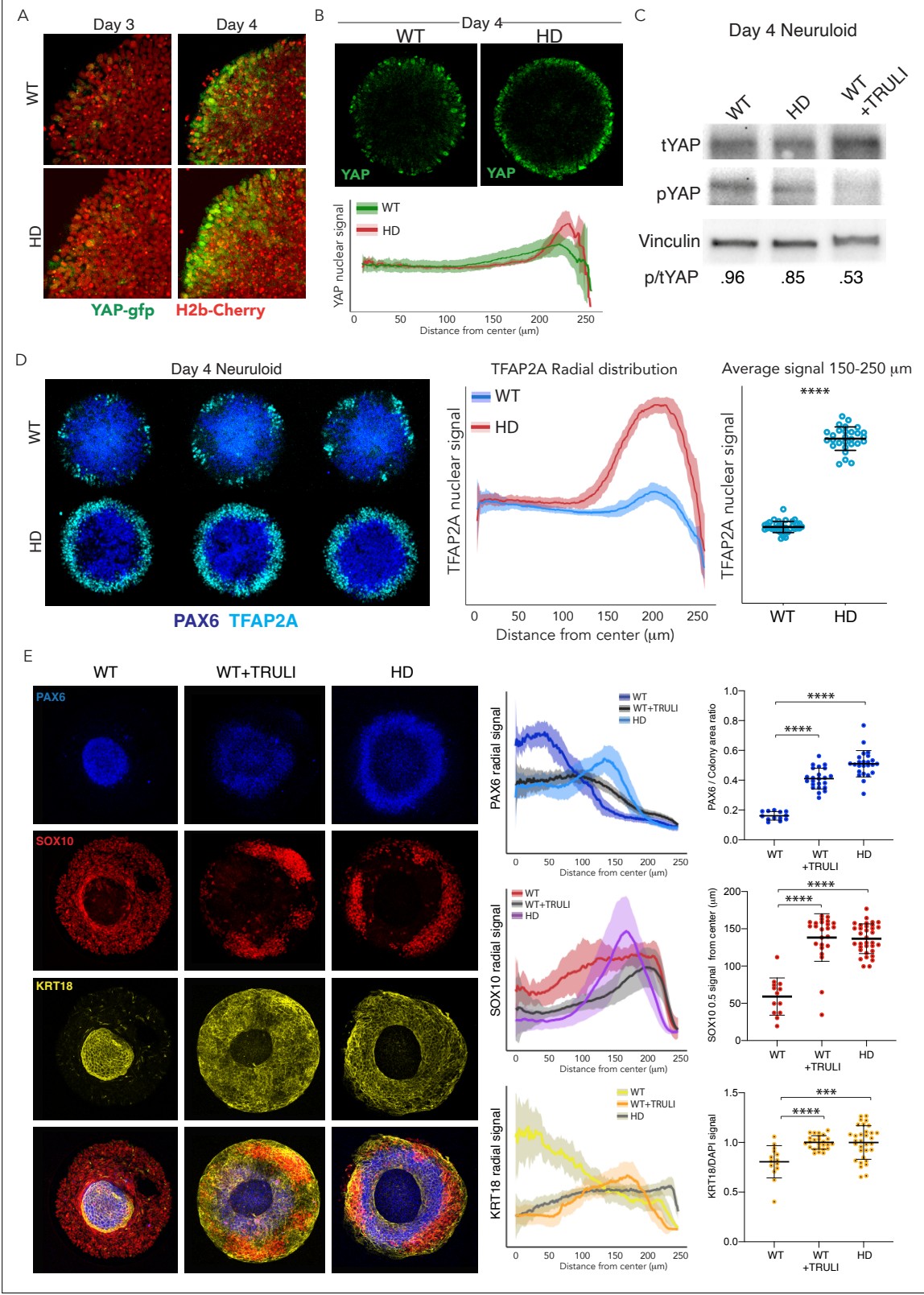

**Figure 4.** Dysregulation of YAP localization in HD neuruloids. (**A**) Fluorescent images of the same portions of living neuruloids before (D3) and 24 hr after BMP4 administration (D4) show increased nuclear YAP in HD than in WT neuruloids. (**B**) Top, YAP immunofluorescence images of D4 neuruloids confirm the effect. Bottom, the radial distribution of YAP nuclear signals is quantitated for 16 neuruloid colonies for each condition. (**C**) Immunoblot analysis of WT and HD neuruloids at D4 shows reduction of YAP phosphorylation on residue S127 (pYAP) in HD colonies, similar to that after TRULI

*Figure 4 continued on next page*

*Figure 4 continued*

treatment of WT colonies. Vinculin provides a loading control. (**D**) Left, Immunofluorescence images of D4 neuroloids demonstrate enhanced TFAP2A at the perimeters of HD colonies. Right, quantitative analysis of nuclear TFAP2A signals confirms the effect. ****, p<0.0001 in unpaired t-tests comparing WT (n=36) and HD (n=28) colonies. (**E**) WT neuroloids (n=13), WT neuroloids treated with 10 μM TRULI (n=23), and HD neuroloids (n=24). Left, representative immunofluorescence images of D7 neuroloids demonstrate the similar phenotypes of TRULI treatment and HD: expansion of NE (PAX6), the diminishment of NC (SOX10), and the enhancement of epidermis (KRT18). Right, quantitative analysis of the three experimental conditions showing radial distributions of the lineage markers PAX6, SOX10, and KRT18 in several colonies. For each colony, plots show the area of the PAX6+ central region as a fraction of the entire colony and the radial distance from the colony's center to the half-maximal intensity of the SOX10 domain. Plots of the KRT18 signal normalized by DAPI labeling confirm the result. ****, p<0.0001 in unpaired t-tests comparing the three experimental conditions. HD, Huntington's Disease.

The online version of this article includes the following video, source data, and figure supplement(s) for figure 4:

**Source data 1.** Raw images for immunoblot shown in *Figure 4C*.

**Figure supplement 1.** Dynamics of YAP localization in early WT and HD neuroloids.

**Figure supplement 2.** Induction of NNE in WT and HD D4 neuroloids.

**Figure supplement 3.** Effect of YAP activation and HD mutation of rate of cell division in neuroloids.

**Figure 4—video 1.** Live YAP dynamics in D3 HD neuroloid upon BMP4 stimulation.

https://elifesciences.org/articles/73075/figures#fig4video1

**Figure 4—video 2.** Live YAP dynamics in D3 HD neuroloid upon BMP4 stimulation (Detail, YAP-GFP and H2b-mCherry).

https://elifesciences.org/articles/73075/figures#fig4video2

**Figure 4—video 3.** Live YAP dynamics in D3 HD neuroloid upon BMP4 stimulation (Detail, YAP-GFP).

https://elifesciences.org/articles/73075/figures#fig4video3

## Neural ectoderm of HD-neuroloids displays ectopic YAP activity

Because we observed elevated YAP activity as early as D4, we characterized this intermediate state by single-cell transcriptomic analysis. The expression profiles of lineage-marker genes in this data set allowed us to annotate the various cell populations (*Figure 5—figure supplement 1A*). As expected, *PAX6* and *TFAP2A* were mutually exclusive and identified NE and NNE, respectively (*Figure 5—figure supplement 1B*). The NNE cell cluster could be subdivided into two subpopulations: NC progenitors expressing *FOXD3* and early epidermis expressing *KRT18*. These populations recapitulated the three major human ectodermal lineages: NE, NC, and epidermis (*Figure 5—figure supplement 1C* and *Figure 5—figure supplement 2*).

Using the cell type-specific markers described above to identify the progenitors of the major ectodermal lineages, we employed scRNA-seq to assess their relative levels of YAP activity. *YAP* and key Hippo pathway components were expressed in all three lineages, most strongly in NC and epidermis (*Figure 5A and B* and *Figure 5—figure supplement 3*). However, the expression levels of YAP target genes, such as CTGF, TAGLN, and TUBB6, were much higher in epidermis than in the other ectodermal lineages (*Figure 5A*). To confirm these observations at a whole-genome scale, we analyzed the expression of 364 known direct YAP targets (*Zanconato et al., 2015*). The majority of these targets were significantly enriched in early epidermal lineage in comparison to other components of D4 neuroloid colonies (*Figure 5B*; *Figure 5—figure supplement 4*). These results confirm our initial observation in WT neuroloids that YAP activity is regulated in the ectodermal lineage and predominantly active in the epidermis lineage (*Figures 2E and 3*, and *Figure 3—figure supplement 1*).

We next used the scRNA-seq data to compare the molecular profiles of D4 early progenitors of NE, NC, and epidermis in WT and HD-neuroloids. Upon performing an analysis of differential gene expression, we ascertained that the majority of YAP target genes were significantly upregulated in HD compared to WT neuroloids (*Figure 5C*). This increased expression of YAP target genes was not observed in the epidermal lineage, for which YAP was highly active in both WT and HD-neuroloids. Instead, YAP targets were ectopically activated in the NC and NE lineages of HD neuroloids by comparison to the WT counterparts (*Figure 5D*). To confirm this observation, we performed gene set-enrichment analyses on a gene set consisting of previously identified YAP target genes (*Zanconato et al., 2015*). In agreement with the analysis of differential gene expression, this approach demonstrated a significant increase in the expression of YAP targets in NC and especially NE lineages of D4 HD neuroloids (*Figure 5E*). At higher resolution, the analysis showed that YAP targets were significantly upregulated in the NE population of D4 HD neuroloids, whereas the upregulation in NNE

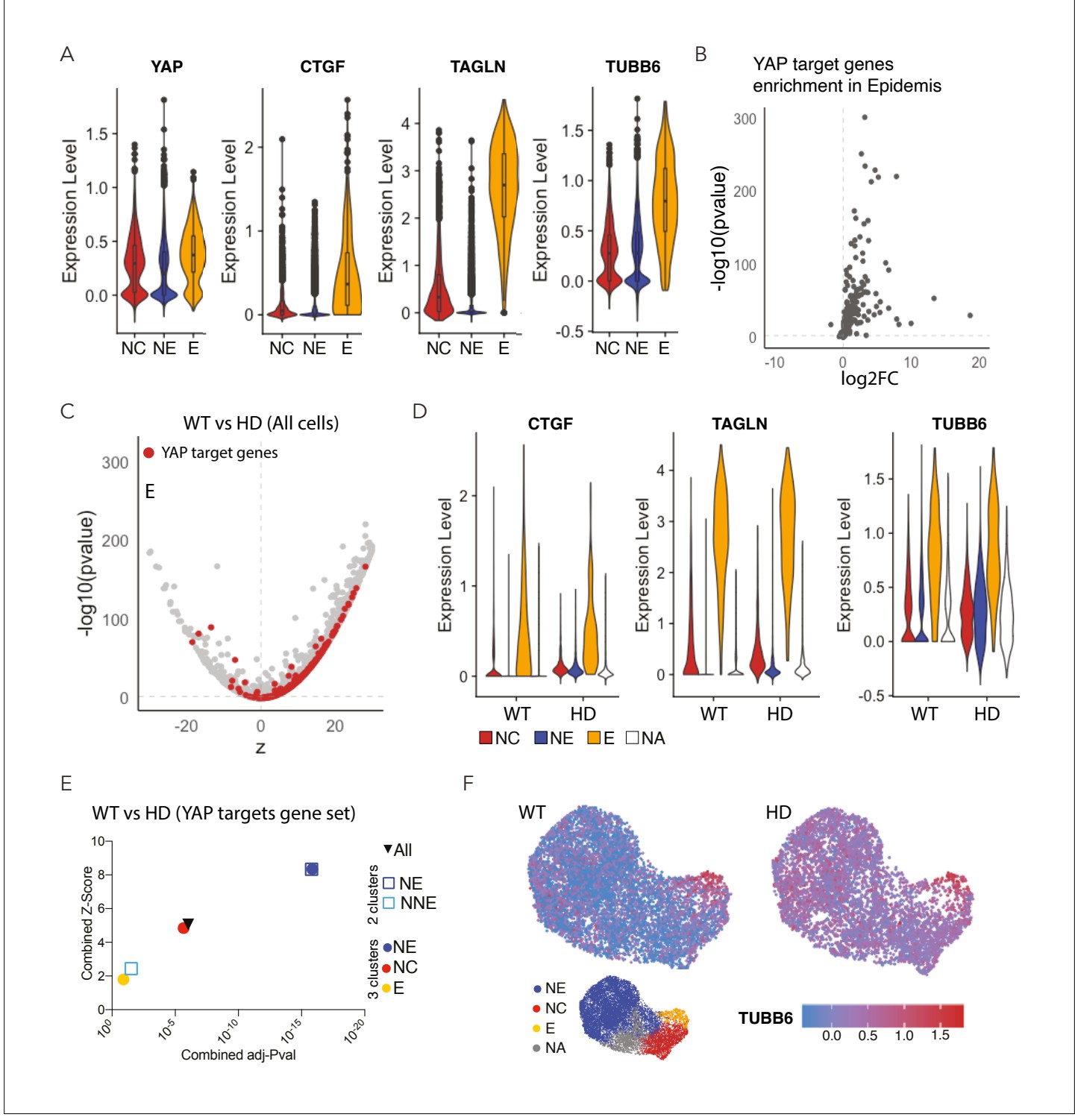

**Figure 5.** Expression of YAP target genes in WT and HD neuruloids. (**A**) Violin plots show the expression of *YAP* and three representative target genes in the three clusters representing the main ectodermal lineages from scRNA-seq analysis of D4 neuruloids. Expression levels are normalized counts, calculated by Seurat for each gene and plotted on a logarithmic scale. (**B**) A volcano plot shows the greater expression of several YAP target genes in epidermis compared to the remainder of the neuruloid. (**C**) An analysis of differential gene expression from scRNA-seq data shows upregulation of YAP target genes (red) in HD with respect to WT neuruloids. (**D**) Violin plots show the expression of three representative YAP target genes in the three lineage clusters from D4 WT and HD neuruloids. The color code is: neural crest (NC), red; neural ectoderm (NE), blue; epidermis (**E**), yellow; and unidentified (NA), white. The augmented expression of YAP target genes is more pronounced in the nonepidermal lineages. (**E**) Gene-set enrichment

*Figure 5 continued on next page*

*Figure 5 continued*

analysis of YAP target genes confirms the effect. (**F**) Analysis by uniform manifold approximation and projection (UMAP) shows the ectopic enhancement of a representative YAP target, *TUBB6*, in D4 WT and HD neuruloids. Full statistics are provided in *Figure 5—figure supplement 5*. HD, Huntington's Disease.

The online version of this article includes the following figure supplement(s) for figure 5:

**Figure supplement 1.** Ectodermal lineage annotation in scRNA-seq from D4 neuruloids.

**Figure supplement 2.** Marker genes for the three main human ectodermal lineages.

**Figure supplement 3.** Expression of key components of the Hippo pathway in D4 neuruloids.

**Figure supplement 4.** Expression of YAP targets in the three human ectodermal lineages at D4.

**Figure supplement 5.** Statistical analysis of D4 neuruloids.

**Figure supplement 6.** Expression of YAP target genes in D7 WT and HD neuruloids.

**Figure supplement 7.** Statistical analysis of D7 neuruloids.

was not significant (*Figure 5E*). Within the NNE lineage, only the NC cluster displayed a significant upregulation of YAP transcriptional activity. This phenomenon was well illustrated by the behavior of the YAP target gene TUBB6, which was normally expressed specifically in the early epidermal lineage but became active throughout the HD neuruloids (*Figure 5F*).

These results confirm that HD neuruloids display higher YAP activity during their self-organization. The data also reveal that such HD-associated activation of YAP does not directly affect the epidermal lineage, in which YAP is endogenously active, but primarily perturbs NE and NC, in which YAP becomes ectopically active as a result of the HD mutation.

## Effects of HD mutation and YAP activation on cell division rate

Because the Hippo pathway is known to regulate the cell cycle (*Totaro et al., 2018*; *Zanconato et al., 2015*), we sought to test whether the hyperactivation of YAP is accompanied by an increase in the rate of cell division in HD neuruloids. We compared the expression of phosphohistone H3 (pH3; J.-Y. *Kim et al., 2017*) in WT and HD neuruloids, in the absence or presence of TRULI, and determined the number of mitotic cells at different times (D4–D7; *Figure 4—figure supplement 3*). Enumeration of the mitotic nuclei in the developing neuruloids revealed that YAP activation by TRULI results in an increased proliferation rate, especially at the latest times (D6 and D7). A comparable but less profound effect appeared in HD neuruloids, which proliferated slightly more than WT cells. This result is consistent with a direct biological effect of YAP hyperactivation in HD neuruloids. The finding is also in agreement with reports showing that human cells carrying CAG expansions at the HTT locus display dysregulated polarity in the cell division of neural progenitors (*Ruzo et al., 2018*; *Zhang et al., 2019*) and that immortalized fibroblasts derived from HD patients show an increase in proliferation (*Hung et al., 2018*).

## The HD phenotype in neuruloids results from ectopic YAP activation

In view of the increased YAP activity in HD-neuruloids and the associated phenotypic consequences, we inquired whether inhibition of YAP transcriptional activity could rescue the HD phenotype. To block YAP activity, we used verteporfin, which prevents the interaction between YAP and TEADs and thereby inhibits the transcription of target genes (*Liu-Chittenden et al., 2012*). In agreement with the observation that a minimal level of YAP activity is required for ectodermal induction (*Giraldez et al., 2021*), administration of verteporfin during the initial 3 days of differentiation resulted in cell death and detachment in both WT and HD samples. In conjunction with BMP4 induction, however, treatment with verteporfin from D3 was not toxic and did not affect the neuruloid phenotype of WT cells (*Figure 6A–C*; *Figure 6—figure supplement 1*). This indicates that high level of YAP activity may be dispensable for epidermal specification.

In D7 HD-neuruloids, administration of verteporfin partially rescued aspects of the HD phenotype. Although the extended epidermal population observed in HD samples was maintained, the inhibition of YAP activity partially restored the NC (SOX10+) lineage and the condensation of the central NE (PAX6+) domain (*Figure 6A–C*; *Figure 6—figure supplement 1*). This result is in agreement with the

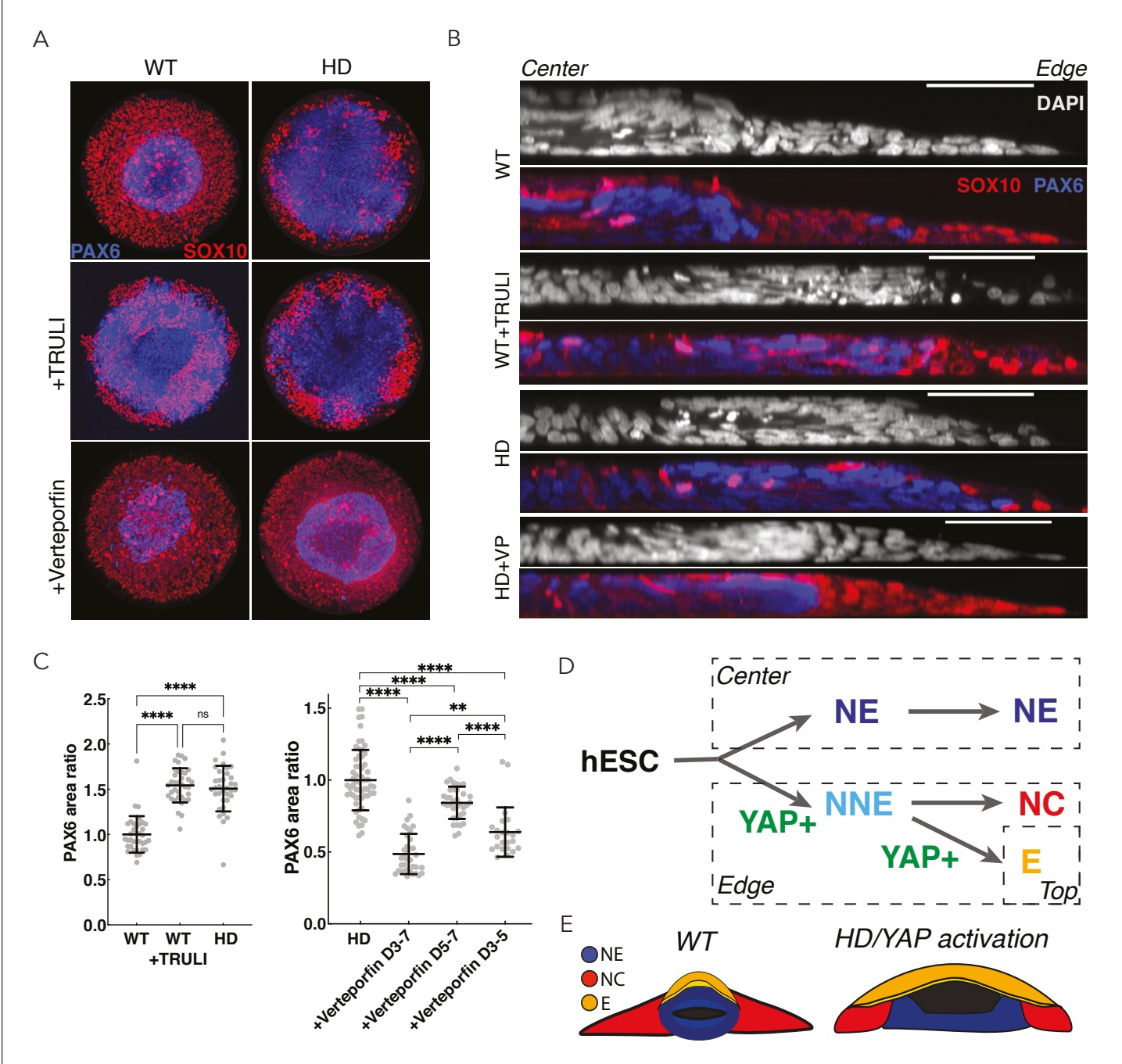

**Figure 6.** Perturbation of YAP activity in WT and HD neuruloids. (**A**) Immunofluorescence images of D7 neuruloids show that the expansion of NE (PAX6, blue) at the expense of NC (SOX10, red) after exposure to 10 μM TRULI or in HD cells. Treatment with 0.3 μM verteporfin has no effect on WT colonies but partially rescues the effect of HD. (**B**) Side views portray neuruloids under the conditions used in (**A**). (**C**) The ratios of PAX6+ areas to control values under various conditions confirm the similarity of TRULI-treated WT neuruloids to HD neuruloids and the suppressive effect of verteporfin (WT colonies, n=36; WT+TRULI, n=36; HD, n=35). Rescue of this HD phenotype is stronger for early than for late exposure to verteporfin (HD, n=60, HD+VP D3–D7, n=35; HD+VP D5–D7, n=36; HD+VP D3–D5, n=25). ****, $p<0.0001$; ns, $p>0.05$ in unpaired t-tests comparing the different experimental conditions. (**D**) In a model of neurulation, YAP activity (green) is posited to favor nonneuronal fates, and hyperactivity (red)—either from TRULI treatment or HD—yields characteristic abnormalities. (**E**) A diagram illustrates the structure of a mature neuruloid and the effects on its architecture of YAP activation or HD mutation. HD, Huntington's Disease; NE, neuronal ectoderm; NNE, nonneuronal ectoderm.

The online version of this article includes the following figure supplement(s) for figure 6:

**Figure supplement 1.** Perturbation of YAP activity in WT and HD neuruloids.

observation that in D4 HD-neuroloids, YAP target genes were significantly upregulated in the precursors of NC and NE, but not in the progenitors of the epidermal lineage (*Figure 5*).

Because YAP activity was elevated during the maturation of HD-neuroloids from D4, we sought to determine the period during which the inhibition of YAP transcriptional activity was most important for rescuing the closure of the central PAX6+ domain. We observed that verteporfin treatment rescued the HD-associated NE expansion if administrated together with BMP4 at D3, whereas it was unable to rescue the phenotype if added after D5 (*Figure 6C*).

These results indicate that the ectopic activity of YAP in HD neuroloids contributes to the malformation of their central PAX6 domains and that the effect is exerted early in neuroloid development.

## Discussion

In numerous organisms and a variety of tissues, the Hippo pathway plays important roles in development, homeostasis, and regeneration (*Moya and Halder, 2019*). Although exploring the role of this pathway during human development remains a challenge, organoid cultures derived from hESCs offer simplified simulacra for investigation. Here, we have identified YAP as an important determinant in neurulation, during which the protein's activity is controlled—at least in part—by the Hippo pathway. Our data suggest that YAP plays a role in the specification of early ectodermal lineages, first influencing the differentiation between NNE and NE, and then between epidermis and NC. Because YAP transcriptional activity is already robust at D4 in precursors of the epidermis, the Hippo pathway might play an early role in the specification of NNE during neurulation (*Figure 6D*).

In addition to the unbalanced differentiation of the NNE, YAP activation also results in the failure of the central NE domain to properly close into a compact structure (*Figure 6E*). Future investigations will determine whether this phenotype is a cell-intrinsic result of the ectopic YAP activation in the NE lineage, or is instead an indirect consequence of the lack of the NC lineage at its margin, which might be required to provide structural support for the closure of the NE domain. This second scenario would be consistent with a recent study showing that both NE and NNE contribute to the folding morphogenesis during human neurulation (*Karzbrun et al., 2021*).

In line with previous reports (*Yao et al., 2014*; *Huang et al., 2016*), our results illustrate a reciprocal relationship between Hippo-YAP and BMP4-SAMD1/5 signaling during the development of the nervous system. In particular, YAP expression rises upon BMP4 stimulation, and YAP activation in turn enhances the cellular response to BMP4 at various stages: initially by augmenting BMP4-dependent NNE induction, and later by favoring the specification of epidermis, which is known to occur at a high BMP4 concentration (*Wilson and Hemmati-Brivanlou, 1995*).

Earlier reports showed that YAP is involved in various stages of NC development (*Zhao et al., 2021*), including the proliferation of NC precursors in animal models such as *Xenopus* (*Gee et al., 2011*). Moreover, YAP has been associated with stem cell-derived NC specification in a BMP4-independent manner in vitro (*Hindley et al., 2016*). These reports contrast with our finding that, in the context of BMP4 induction, excessive YAP activity can skew NNE specification toward an epidermal lineage at the expense of NC. These differences might stem from an epistatic relationship between the Hippo-YAP and BMP pathways during human neurulation, when YAP activity could sensitize cells to BMP stimuli.

In agreement with recent reports showing that YAP is required for early ectodermal specification (*Giraldez et al., 2021*), the pharmacological inhibition of YAP transcriptional activity with verteporfin during the early phase of neural induction (D0–D3) results in cell death. However, subsequent administration of verteporfin is inconsequential for the generation of WT neuroloids (*Figure 6*; *Figure 6—figure supplement 1*). This result indicates that YAP can influence ectodermal specification during neurulation, perhaps by boosting BMP4 signaling, but its activity is not required.

Noting the similarities between TRULI-treated neuroloids and those bearing HD mutations (*Haremaki et al., 2019*), we explored the potential contribution of YAP dysregulation to the HD phenotype. HD-neuroloids displayed altered YAP regulation that resulted in ectopic activation of its transcriptional program, especially within the NE lineage. This HD-associated YAP hyperactivity results in an unbalanced differentiation of the NNE lineage as well as increased cell proliferation and might contribute directly to the failure of compaction in the central NE, even though this structure is devoid of YAP detectable by immunofluorescence.

Although individuals with HD have not been reported to experience a failure in folding of the neural tube or effects on fertility rate, there is a large body of evidence that connects HTT activity to developmental events. First, homozygote loss of function in the mouse leads to embryonic lethality at the onset of gastrulation, demonstrating that the function of the protein is necessary for early embryogenesis (*Nasir et al., 1995*). Consistent with that result, no human of the HTT$^{-/-}$ genotype is ever born. Second, in human embryos, the HD mutation impacts development as early as gestational week 13 (*Barnat et al., 2020*). Moreover, CAG expansion in HD patients causes early fetal anatomical abnormalities including an effect on ventricular volume (*Hobbs et al., 2010*). The earliest postnatal death due to HD in humans has been reported at 18 months of age (*Nicolas et al., 2011*). Perhaps the most stringent evidence linking the causality of HD to early embryogenesis is the fact that a pulse of expression of expanded CAG-HTT in a WT mouse leads to the emergence of HD symptoms in the adult (*Molero et al., 2016*; *Wiatr et al., 2018*). Third, guided differentiation of hESCs toward mature cortical neurons, self-organizing cerebroids, gastruloids, and neuruloids all point to CAG-length-dependent embryonic phenotypes (*Galgoczi et al., 2021*; *Haremaki et al., 2019*; *Ruzo et al., 2018*). Our study demonstrates that HTT-CAG expansion affects a model for human neurulation and that this phenotype is associated with the dysregulation of the Hippo pathway, with an increase of YAP activity early at D4 and later reduction by D7.

Owing to homeostatic and compensatory mechanisms, development in vivo is likely to be more robust than that in vitro. As a result, the neuruloid offers a sensitive assay for subtle phenomena that might otherwise remain hidden. Because our data suggest that neurulation is perturbed in HD, exploring how the developing human embryo compensates for this challenge could yield insight into the dysfunction manifested postnatally.

Several studies support the role for YAP in the pathophysiology of HD. Postmortem cortical samples from HD patients display diminished YAP activity. Moreover, YAP activators ameliorate symptoms in HD mice and normalize the phenotypes of cultured cells (*Mao et al., 2016*; *Mueller et al., 2018*). In our data, YAP activation at D4 was elevated in HD neuruloids compared to WT controls, but at D7, YAP transcription was instead diminished (*Figure 5—figure supplement 6*). This reversal might reflect compensation for increased YAP activity so that development could proceed normally. The reversal might alternatively represent an end state of failed development. It will be interesting to investigate whether YAP hyperactivation during early neurulation leads to hypoactivity in mature neurons. Lats1/2 knockout mice demonstrate that Hippo signaling later regulates the number and differentiation of neural progenitor (*Lavado et al., 2018*), and postmortem analysis of HD patients shows significantly increased proliferation of subependymal neural progenitors (*Curtis et al., 2005*). These results suggest that the effects of *HTT* mutations on this process through YAP dysregulation bear on the pathophysiology of the disease.

Our study leaves open the question of the connection between Hippo signaling and HD. Huntingtin protein is known to play a role in vesicle trafficking and establishing apical-basal polarity (*Caviston and Holzbaur, 2009*; *Galgoczi et al., 2021*; *Saudou and Humbert, 2016*), a prime regulator of Hippo signaling (*Piccolo et al., 2014*). The protein is also involved in the establishment and maintenance of intact epithelia, a function that is compromised by HD mutations (*Galgoczi et al., 2021*). Loss of intercellular adhesion might contribute directly to dysregulation of the Hippo pathway, reflected here as increased YAP activity. In this scenario, it is possible that CAG expansion at the HTT locus effects cell-cell contact, disrupting proper epithelial integrity and in turn dysregulating the Hippo pathway. Furthermore, lack of proper epithelial polarity could also result in the mislocalization of morphogen receptors and thus in increased sensitivity of BMP4/SMAD1/5 signaling, which in turn could affect the Hippo pathway. Future investigations of these mechanisms will reveal the molecular consequences of HD mutations and how mechanical cues interact with morphogenic signaling during embryonic development.

## Materials and methods
### Cell line
This work was based on the RUES2 (National Institutes of Health Human Embryonic Stem Cell Registry #0013) human ES cell line, the identity of which was confirmed by STR profiling (performed by Cell Line Genetics). The cells were also tested for mycoplasma and resulted negative.

## Cell culture

Cells were grown in a HUESM medium that was conditioned with mouse embryonic fibroblasts and supplemented with 20 ng/ml bFGF (MEF-CM; *Deglincerti et al., 2016*). Cells were grown on tissue culture dishes coated with Geltrex solution (Life Technologies) and tested for mycoplasma at 2-month intervals.

## Generation of YAP-GFP hESC lines

Carboxy-terminal tagging of endogenous YAP alleles with GFP was obtained by using a donor plasmid, which was kindly donated by the Liphardt laboratory (*Franklin et al., 2020*). This contains an upstream homology arm approximately 1 kB in length; a cassette encoding GFP, a P2A site, puromycin resistance, and a stop codon; and a downstream homology arm also about 1 kB long. RUES2 (WT) and isogenic 56CAG (HD) hESC lines were co-transfected with Px330-Cas9-sgYAP (sgRNA sequence targeting YAP locus: TTAGAATTCAGTCTGCCTGA), donor YAP-eGFP-P2A-puromycin resistance, ePB-H2B-mCherry-BSDa, and transposase. Nucleofection was performed with program B-016 on an Amaxa Nucleofector II (Lonza). Cells were then grown for 10 days under selection by 1 µg/ml puromycin and 10 µg/ml blasticidin. Multiple clones from each line were selected for characterization by PCR genotyping with the following oligonucleotides:

> GFP integration:
> YAP to exon9-1F: CAGGGGTAATTACGGAAGCA
> mEGFP_R: CTGAACTTGTGGCCGTTTAC.
> Heterozygotic versus homozygotic:
> Yap Het/Homo check LHA F: GGTGATACTATCAACCAAAGCACCC.
> YAP Het/Homo check R: CATCCATCATCCAAACAGGCTCAC.

## Neuruloid culture

To generate neuruloids (*Haremaki et al., 2019*), we coated micropatterned glass coverslips (CYTOO Arena A, Arena 500 A, and Arena EMB A) for 3 hr at 37°C with 10 µg/ml recombinant laminin-521 (BioLamina, LN521-05) diluted in PBS+/+ (Gibco). Single-cell suspensions were then incubated for 3 hr at 37°C in HUESM medium supplemented with 20 ng/ml bFGF and 10 µM Rock inhibitor Y27632. The micropattern culture was then washed once with PBS+/+ and incubated with HUESM with 10 µM SB431542 and 0.2 µM LDN 193189. On D3, the medium was replaced with HUESM containing 10 µM SB431542 and 3 ng/ml BMP4. On D5, the medium was replaced with the same fresh medium and then incubated until D7.

## NC and epidermis differentiation

Direct differentiation of NC was performed by a published method (*Tchieu et al., 2017*). RUES2 hESCs were cultured for 3 days in E6 medium (STEMCELL, 05946) containing 1 ng/ml BMP4, 10 µM SB431542, and 600 nM CHIR, then transferred for 4 days to 10 µM SB431542 and 1.5 µM CHIR.

Induction of epidermis was conducted on Transwell (Corning, CLS3413). RUES2 hESCs were cultured for 3 days in E6 medium (STEMCELL, 05946) containing 10 µM SB431542 (top of the transwell) and 10 µM SB431542 and 10 ng/ml BMP4 (bottom of the transwell). After the media had been replaced with 10 µM SB431542 and 0.2 µM LDN193189 (top and bottom), the cells were incubated for an additional 4 days.

## Immunofluorescence

Micropatterned coverslips were fixed with 4% paraformaldehyde (Electron Microscopy Sciences 15713) in warm medium for 30 min, rinsed three times with PBS−/−, and blocked and permeabilized for 30 min with 3% normal donkey serum (Jackson ImmunoResearch 017-000-121) and 0.5% Triton X-100 (Sigma-Aldrich 93443) in PBS−/−. Specimens were incubated with primary antibodies for 1.5 hr at room temperature, washed three times for 5 min each in PBS−/−, incubated with secondary antibodies conjugated with Alexa 488, Alexa 555, Alexa 594, or Alexa 647 (Molecular Probes) at 1/1000 dilution. After 30 min incubation with 100 ng/ml 4′,6-diamidino-2-phenylindole (DAPI; Thermo Fisher Scientific D1306), specimens were washed three times with PBS−/−. Coverslips were mounted on slides using ProLong Gold antifade mounting medium (Molecular Probes P36934).

Antibodies used:
YAP (Santa Cruz Biotechnology 101199);
TFAP2A (DSHB 3B5 concentrate);
PAX6 (BD Biosciences 561462);
SOX10 (R&D Systems AF2864); and
KRT18 (Abcam ab194130).

## Imaging and analysis

Confocal images were acquired on a Zeiss Inverted LSM 780 laser-scanning confocal microscope with a 10×, 20×, or 40× water-immersion objective. YAP-GFP Live reporter imaging was performed with a Zeiss AxioObserver z1 or spinning-disk microscope (Cell Voyager CV1000, Yokogawa).

For the acquisition of radial profiles, images were preprocessed to display the maximum-intensity projection (MIP) of at least four z-stacks. MIP images were subsequently imported into Python (czifile 2019.7.2). Individual colonies were cropped, and the radial intensity profile of fluorescence was calculated for each image starting from the center. Results were reported in arbitrary fluorescence units.

For quantitative analysis of single cells (*Figure 2D*), individual nuclei were segmented in the DAPI channel in each z-slice of a confocal image by a published procedure (*Etoc et al., 2016*). Twelve images representative of the data were selected to train Ilastik, an interactive learning and segmentation toolkit, to classify each pixel into two categories: nucleus or background. The pixels in the nucleus class were then used to create a binary mask. The original z-slice image was then processed to detect seeds based on a sphere filter with a manually set threshold. Watershedding was then applied from the defined seeds into the mask of the nuclei as defined previously by Ilastik, resulting in a set of segmented nuclei. The median intensity of PAX6, SOX10, or YAP immunofluorescence was then measured for each segmented nucleus.

## Single-cell RNA sequencing

Micropatterned glass coverslips with neuruloids 500 μm in diameter were grown in 3 ng/ml BMP4 from RUES2 hESCs and the isogenic 56CAG HD line. On D4, approximately half of the neuruloids on each coverslip were scraped off and treated for 10 min at 37°C with Accutase (Stemcell Technologies). The remaining neuruloids were fixed for immunofluorescence analysis as a quality control. After dissociation, the cells were washed three times in PBS−/− (Gibco) with 0.04% bovine serum albumin and strained through a 40-μm tip strainer (Flowmi 136800040). The number and viability of cells were determined by exclusion of trypan blue on a Countess II automated cell counter. The samples were separately loaded for capture with the Chromium System using Single Cell 3' v2 reagents (10× Genomics). Following cell capture and lysis, cDNA was synthesized and amplified according to the manufacturer's instructions. The resulting libraries were sequenced on NovaSeq 6000 SP flowcell. The Cell Ranger software pipeline (v2.0.2) was used to create FASTQ files that aligned to the hg19 genome with default parameters. Filtered gene-expression matrices of genes versus detected barcodes (cells) with counts of unique molecular identifiers were generated and used for subsequent analyses. These data are available through NCBI accession number GSE182074.

## Single-cell RNA analysis

The sequencing data were analyzed using Cellranger count (v3.0.2) and aggregated with the Aggr function and default settings. The aggregated datasets were processed with Seurat (v3.2.3) (*Stuart et al., 2019*). Quality control and filtering were performed with Scater Bioconductor (v1.16.1) to identify and remove poor-quality cells characterized by low library sizes, low numbers of expressed genes, and low or high numbers of mitochondrial reads (*McCarthy et al., 2017*). Following a cutoff step (counts >6000; genes >2600; 3% <mitochondrial genes <11 %), we obtained about 8000 cells in WT and roughly 6000 cells for HD. Counts of genes in each cell were normalized and $\log_{10}$-transformed. In order to merge the data sets using Seurat's FindIntegrationAnchors and IntegrateData methods, we identified integration anchors for the WT and HD datasets by means of the top 2000 variable genes and first 20 principal components. The cell cycle phase was estimated by Seurat with default setting and data were rescaled to remove the effects of cell cycle and mitochondrial transcript expression. Clustering of cell was performed with Seurat's findClusters function by use of the first 20 principal components and with two resolutions—0.5 and 0.1—to yield sets of seven clusters and two

clusters, respectively. The seven-cluster set was further aggregated by manual curation of cell-type specific markers into four clusters. Uniform manifold approximation and projection (UMAP) dimension reduction implemented within Seurat was applied to the normalized, integrated data and the cluster sets projected and visualized on the resulting UMAP coordinates. Visualization of normalized expression values on UMAP coordinates and as violin plots were performed using Seurat's FeaturePlot and VlnPlot, respectively. Differential gene expression and analysis between clusters and samples were performed by MAST (v1.8.2; *Finak et al., 2015*). YAP targets were retrieved from msigDB C6 gene sets and enrichment with differential expression tests assessed by single-cell GSEA implemented within MAST.

## Immunoblotting

Protein samples were resolved in 4%–15% Mini-PROTEAN TGX Stain-Free Gels (for HTT: NuPAGE 3%–8% Tris-Acetate Protein Gels), then transferred to PVDF membranes and incubated with primary antibodies overnight at 4°C followed by secondary antibodies at RT for 1 hr. Pierce ECL Western Blotting Substrate or SuperSignal West Femto Maximum Sensitivity Substrate (Thermo Fisher Scientific) was used for the detection of peroxidase activity.

> Antibodies used:
> Vinculin (MilliporeSigma 05-386);
> GFP (Thermo Fisher Scientific A 11122)
> tYAP (Santa Cruz Biotechnology sc 101199)
> phYAP (CST 4911 S); and
> HTT (MilliporeSigma mab2166).

## Acknowledgements

FMP, TH, QT, TLL, AJB, and AHB were supported by the CHDI foundation (A-9423). NRK was supported by NIGMS Grant T32GM007739. TLL was also supported by FCT PD/BD/127997/2016. RDS was supported by EMBO-LTF-254-2019. KG was supported by NIDCD Grant R21DC016984. AJH is an Investigator of Howard Hughes Medical Institute.

## Additional information

### Competing interests

Nathaniel R Kastan, Ksenia Gnedeva: is part to an application for patent protection of derivatives of the Lats inhibitor TRULI used in this study. Fred Etoc: is a shareholder of RUMI Scientific. AJ Hudspeth: Is part to to an application for patent protection of derivatives of the Lats inhibitor TRULI used in this study. Ali H Brivanlou: is a co-founder and shareholder of RUMI Scientific. The other authors declare that no competing interests exist.

### Funding

| Funder | Grant reference number | Author |
|---|---|---|
| CHDI Foundation | A-9423 | Francesco M Piccolo<br>Tomomi Haremaki<br>Qingyun Tian<br>Tiago L Laundos<br>Andrew J Beaudoin<br>Ali H Brivanlou |
| National Institute of General Medical Sciences | T32GM007739 | Andrew J Beaudoin |
| European Molecular Biology Organization | EMBO-LTF-254-2019 | Riccardo De Santis |
| National Institute on Deafness and Other Communication Disorders | R21DC016984 | Ksenia Gnedeva |

| Funder | Grant reference number | Author |
|---|---|---|
| Howard Hughes Medical Institute | | AJ Hudspeth |
| Fundação para a Ciência e a Tecnologia | PD/BD/127997/2016 | Tiago L Laundos |

The funders had no role in study design, data collection and interpretation, or the decision to submit the work for publication.

## Author contributions

Francesco M Piccolo, Conceptualization, Data curation, Formal analysis, Visualization, Writing – original draft; Nathaniel R Kastan, Resources, Writing – original draft; Tomomi Haremaki, Investigation, Methodology; Qingyun Tian, Tiago L Laundos, Investigation; Riccardo De Santis, Software; Andrew J Beaudoin, Thomas S Carroll, Ji-Dung Luo, Data curation; Ksenia Gnedeva, Resources; Fred Etoc, Conceptualization, Methodology, Software; AJ Hudspeth, Ali H Brivanlou, Supervision, Writing – review and editing

## Author ORCIDs

Francesco M Piccolo ⓘ http://orcid.org/0000-0002-5491-1454
Nathaniel R Kastan ⓘ http://orcid.org/0000-0002-7044-9685
Tiago L Laundos ⓘ http://orcid.org/0000-0002-5727-565X
Ji-Dung Luo ⓘ http://orcid.org/0000-0003-0150-1440
Ksenia Gnedeva ⓘ http://orcid.org/0000-0002-3870-9256
AJ Hudspeth ⓘ http://orcid.org/0000-0002-0295-1323
Ali H Brivanlou ⓘ http://orcid.org/0000-0002-1761-280X

## Decision letter and Author response

Decision letter https://doi.org/10.7554/eLife.73075.sa1
Author response https://doi.org/10.7554/eLife.73075.sa2

# Additional files

## Supplementary files

• Transparent reporting form

## Data availability

Sequencing data have been deposited in GEO under accession code GSE182074.

The following dataset was generated:

| Author(s) | Year | Dataset title | Dataset URL | Database and Identifier |
|---|---|---|---|---|
| Piccolo FM, De Santis R, Brivanlou AH, Luo J, Carroll TS | 2022 | scRNAseq profile of Day 4 Neurloids derived from WT and HD isogenic hESC lines | http://www.ncbi.nlm.nih.gov/geo/query/acc.cgi?acc=GSE182074 | NCBI Gene Expression Omnibus, GSE182074 |

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
