## [Editor Report]

This manuscript harnesses an organoid model of human neurulation to unravel the role of the Hippo signalling pathway in the specification of the three key ectodermal cell types. The authors then investigate how these mechanisms are dysregulated in an organoid model of Huntington's Disease. This work will be of broad interest to readers interested in the process of neurulation and how dysregulation of developmental mechanisms may lead to disease conditions in adulthood.

---

## [Decision Letter]

**Decision letter after peer review:**

Thank you for submitting your article "Role of YAP in early ectodermal specification and a Huntington's Disease model of human neurulation" for consideration by *eLife*. Your article has been reviewed by 3 peer reviewers, and the evaluation has been overseen by a Reviewing Editor and Marianne Bronner as the Senior Editor. The following individual involved in review of your submission has agreed to reveal their identity: Raman M Das (Reviewer #3).

The reviewers have discussed their reviews with one another, and the Reviewing Editor has drafted this to help you prepare a revised submission. While the reviewers see merit in the work, they also recommend extensive revisions to improve quantitation and clarify the results. The full reviews are enclosed to help in the revision process.

Essential revisions:

1. The manuscript requires additional quantifications and an expanded discussion.

2. The authors should look at upstream effectors of the Hippo pathway and assess the effects of YAP activity on proliferation.

3. The authors must resolve the inconsistencies in the model and improve quantification strategies, as detailed in the full reviews below.

*Reviewer #1 (Recommendations for the authors):*

– In Figure 5 the authors present the upregulation of YAP target genes in the epidermal lineage. The dataset of 364 genes employed for this analysis (Zanconato et al., 2015) was developed in epithelial-like breast cancer cells and may not represent the downstream targets of Hippo signaling in the developing ectoderm.

– Introduction and discussion are insufficiently referenced.

– Statistics for violin plots in figures 5, S6, S7, S9 and for box plots in Figure 2D should be included.

– Legends of Fig6 and FigS8 indicate "PAX10" expression in NC instead of SOX10.

– Supplemental figure 2A and 2B do not correspond to the description in the figure and text.

*Reviewer #2 (Recommendations for the authors):*

This manuscript identifies YAP signalling as key player in lineage determination during development of early human ectoderm. Additionally, the authors show that neuroloids generated using cells engineered to express penetrant levels of CAG repeats in the HTT gene display aberrant YAP signalling during ectodermal specification and that this phenotype can be partially rescued by inhibition of this pathway. The similarity of the YAP-activated neuroloids and the HD neuroloids is striking and important. However, the study should be improved by providing clear mechanistic experiments to definitively demonstrate the role of YAP signalling in NNE specification and in HD neuroloids.

– It could also be useful to look at various proteins within the Hippo pathway to show dynamics across the colony over neural induction.

– In figure 1E, it is hard to distinguish the localization of YAP within the nucleus – it would be useful to show higher resolution images with H2B-Cherry cells like in SFig2, or simply fixed-and-stained for YAP compared to DAPI to show redistribution of YAP over neural induction. Quantification of colocalization within the nucleus is needed.

– It would be helpful if the authors could show higher resolution still images to document what they mean by "nuclear cytoplasmic flux." Right now the data is rather unconvincing that it is shuttling out of the nucleus. Quantification of fluorescence intensities within the nucleus compared to outside of the cell would be needed to show that YAP is indeed shuttling outside the nucleus. Would it be possible to use a cell membrane marker to show that this YAP shuttling conserves the protein over time – as in, all of the accounted for nuclear YAP is maintained and pushed to the cytoplasm?

– Quantifications of fluorescence intensities of the blots would be useful to measure increases in expression.

– Statistics are needed for the change in expression levels in 2C – is this a significant change? Also what are the Y axes?

– In 2D they describe nuclear YAP exclusion from the nucleus but it is difficult to discern where exactly the signal is coming from without a DAPI channel or zoomed images of nuclear exclusion. It would be also interesting to describe the extent of nuclear exclusion and if that correlates with KRT18 expression. Does the more nuclear YAP dictate how much KRT18 expression there is?

– Similar quantifications of cytoplasmic/nuclear expression is needed for Figure 4.

– The authors describe that verteporfin results in loss of cell adhesion in both WT and HD samples but do not show images describing this phenomenon. As a control it would be useful to show that adhesions are maintained over the course of neuroloid development with and without Verteporfin treatment.

Perhaps most importantly, the authors demonstrate that the YAP signature is highly enriched in the NE cluster in HD neuroloids, however, their staining and previous figures show little to no YAP localisation within the PAX6+ NE.

Significant extra analysis of the downstream ramifications of aberrantly increased YAP signalling in the NE is necessary.

The authors claim that increased YAP activity leads to increased NNE and epidermal cell commitment but this is not reflected in the scRNAseq data (Figure 5). It would be valuable to include more robust identification of lineages that includes multiple markers for each lineage (a heat map will illustrate this nicely).

– While the authors use TRULI and Verteporfin to study the effects of YAP on neural induction, it would be worthwhile to visualize via fix-and-stains other members of the HIPPO pathway (and especially members downstream of YAP) to show if there are similar effects of nuclear exclusion in WT and HD neuruloids.

*Reviewer #3 (Recommendations for the authors):*

The authors report exclusion of Yap from nuclei at D3, when neural induction occurs. Is this indicative of initiation of Hippo signalling? Are the upstream stimuli that may lead to Hippo activation here known? Perhaps the authors could speculate on this a bit more in the discussion? Also, YAP is again upregulated upon BMP stimulation at D4, but is BMP directly activating Hippo signalling here?

Although of considerable interest, the connection between human neurulation and Huntington's Disease could be improved a bit. I wonder if the authors could expand on their discussion of the similarities between TRULI treatment and HD neuruloids. This is a particularly intriguing finding, as the authors state, since HD symptoms manifest only in adulthood. Are embryos carrying the htt mutation less likely to complete development and is there an effect in fertility rates? It would also be interesting if the authors could comment on juvenile HD, which has a much earlier onset of symptoms.

Specific comments:

1. Overall, the authors have not presented key experimental details such as the number of neuruloids assessed for each experiment. This information will need to be added to improve confidence in the conclusions.

2. Perhaps the presentation of Video 1 could be improved to better display the nuclear-cytoplasmic flux of YAP. An enlarged region of representative cells undergoing said flux would be a welcome addition here as would slowing down the video.

3. The data presented in supplementary figure 2 is meant to show the effect of cell density and TRULI treatment on YAP localisation. However, this figure is difficult to interpret as all the panes appear to have similar YAP expression and localisation. Has this data been quantified? An indication of the percentages of cells with nuclear or cytoplasmic YAP would be useful here as would be zoomed images of representative cells.

4. Figure 2D is difficult to interpret as presented here. For example, it is not clear if any Pax6 positive cells are also expressing YAP. It would be useful to split the channels here.

5. Although it is clear that YAP is present in KRT18 nuclei (Figure 2E), there are a number of other cells in the region that are KRT18 negative but still present nuclear YAP. Are these cells also destined to acquire epidermal fate?

6. On page 6 and in following sections, the authors state that administration of TRULI leads to enhanced YAP activity. However, the data only really demonstrates nuclear localisation rather than activity. Granted, the later scRNA seq experiments then confirm this finding. Perhaps this statement could be toned down here.

7. The control images shown in figure 3 appear to also possess a lumen, albeit this is smaller than the lumen of the TRULI treated neuruloids. Is this indicative of slower neurulation compared to controls? Should neurulation be complete at the stages shown? I assume this is the case as the images shown in figure 2D show fully closed lumens.

8. The authors report that administration of verteporfin in the first three days of culture leads to loss of cell adhesion, but there is not data shown to support this.

9. The methods section should also include a section on immunoblotting.

---

## [Author Response]

Essential revisions:1. The manuscript requires additional quantifications and an expanded discussion.

We have now added quantification of nuclear YAP accumulation and included additional quantification of live YAP-GFP in pluripotency. The discussion has also been expanded to include a potential connection between the HD-neuruloid phenotype and the pathophysiology of HD.

2. The authors should look at upstream effectors of the Hippo pathway and assess the effects of YAP activity on proliferation.

We have addressed below in our response to reviewers, our analysis of upstream components derived from our data that indicate expression of all canonical components of the Hippo pathway at comparable expression levels across ectodermal lineages and between WT and HD samples. We also expand on our hypothesis regarding the connection between HD and YAP dysregulation as well as current technical limitations in further speculations regarding the upstream effectors of the Hippo pathway. We have also assessed, by two independent means the effects of YAP activation on the rate of cell division, using our scRNA-seq dataset at D4 to calculate the velocity of cell cycle-associated genes, and a direct count of mitotic cells at different time points during neuruloid formation. This analysis is now presented in a new supplemental figure (Figure 4—figure supplement 3). We show that there is an acceleration of the cell cycle associated with YAP activation or HD mutation, reinforcing our main conclusion that HD neuruloids display hyperactive YAP.

3. The authors must resolve the inconsistencies in the model and improve quantification strategies, as detailed in the full reviews below.

We have now resolved apparent inconsistencies in our model by improving our schematic on Figure 6E, and provide further clarifications throughout the manuscript. We have also added nine supplemental figures that we believe, strengthen the conclusions of our study. Quantification strategies and methods have been clarified and strengthened as requested.

Reviewer #1 (Recommendations for the authors):– In Figure 5 the authors present the upregulation of YAP target genes in the epidermal lineage. The dataset of 364 genes employed for this analysis (Zanconato et al., 2015) was developed in epithelial-like breast cancer cells and may not represent the downstream targets of Hippo signaling in the developing ectoderm.

The reviewer is correct, the list of YAP target was established in a different context, which is not necessarily maintained here. However, we believe that this is the best approximation available to determine genes that are potentially directly regulated by YAP. Moreover, commonly cited YAP target genes are included in the dataset, which is also used in other studies (Kastan et al., 2021). Given epigenetic accessibility, the target specificity for a transcription factor is given by the promoter sequences, which do not change in different contexts. Therefore, it is true that not all 364 genes are YAP targets in this neuruloid context, but many of them are in progenitors of the epidermis lineage, which is the one that displays the strongest nuclear YAP signals.

– Introduction and discussion are insufficiently referenced.

We now added additional references to introduction and Discussion.

– Statistics for violin plots in figures 5, S6, S7, S9 and for box plots in Figure 2D should be included.

We would like to thank the reviewer for pointing this out. We have now included additional supplemental figures containing tables for the statistical analysis of scRNA-seq violin plots (Figure 5 and Figure 5—figure supplement 1 and Figure 5—figure supplement 17). Furthermore, we now include relevant statistics for the boxplot in Figure 2D.

– Legends of Fig6 and FigS8 indicate "PAX10" expression in NC instead of SOX10.

Thank you for pointing this out; we have corrected the typo.

– Supplemental figure 2A and 2B do not correspond to the description in the figure and text.

Thank you for pointing this out; we have now corrected this discrepancy.

Reviewer #2 (Recommendations for the authors):This manuscript identifies YAP signalling as key player in lineage determination during development of early human ectoderm. Additionally, the authors show that neuroloids generated using cells engineered to express penetrant levels of CAG repeats in the HTT gene display aberrant YAP signalling during ectodermal specification and that this phenotype can be partially rescued by inhibition of this pathway. The similarity of the YAP-activated neuroloids and the HD neuroloids is striking and important. However, the study should be improved by providing clear mechanistic experiments to definitively demonstrate the role of YAP signalling in NNE specification and in HD neuroloids.– It could also be useful to look at various proteins within the Hippo pathway to show dynamics across the colony over neural induction.

We have expanded the discussion about potential consequences of increased YAP activity and looked into the expression level of other components of the Hippo pathway across the three ectodermal lineage progenitors at D4 (Figure 5—figure supplement 3). This analysis showed that the Hippo pathway is active in all the cells at this stage. Unfortunately, antibodies for the proteins expressed by these genes did not work in our hands.

– In figure 1E, it is hard to distinguish the localization of YAP within the nucleus – it would be useful to show higher resolution images with H2B-Cherry cells like in SFig2, or simply fixed-and-stained for YAP compared to DAPI to show redistribution of YAP over neural induction. Quantification of colocalization within the nucleus is needed.

Figure 1E correspond to samples that were fixed and labeled for YAP, TFAP2A and DAPI. The DAPI signal was then used to segment individual nuclei, which allowed the measurement of the YAP signal in the nucleus of each cell. Following the reviewer’s recommendation, and to better clarify this point we now include magnified videos of live YAP-GFP;H2B-mCHerry and YAP-GFP neuruloids (D3-D4; Figure 1—video 2, Figure 1—video 3, Figure 1—video 5, Figure 1—video 6, Figure 4—video 2 and Figure 4—video 3); as well as magnified confocal images of fixed and stained (DAPI/YAP) D4 neuruloids, in both cases for WT, WT+TRULI, and HD lines (Figure 1—figure supplement 5).

– It would be helpful if the authors could show higher resolution still images to document what they mean by "nuclear cytoplasmic flux." Right now the data is rather unconvincing that it is shuttling out of the nucleus. Quantification of fluorescence intensities within the nucleus compared to outside of the cell would be needed to show that YAP is indeed shuttling outside the nucleus. Would it be possible to use a cell membrane marker to show that this YAP shuttling conserves the protein over time – as in, all of the accounted for nuclear YAP is maintained and pushed to the cytoplasm?

We apologize for the confusion from our use of the term “flux". Upon BMP4 induction, YAP is upregulated and found in the nucleus of some cell at the edge of the colony. This effect is regulated by the Hippo pathway, which when active—in terms of phosphorylation—retains YAP in the cytoplasm. Thus, TRULI (a Hippo inhibitor) increases the amount of YAP that becomes free to translocate into the nucleus (Figure 1E and Figure 1—figure supplement 5). Our focus is not with the bidirectional flux but rather the absolute measurable *increase* in nuclear localization upon Hippo inhibition.

We performed these analyses using confocal images of cells labeled for YAP and DAPI, using the latter to identify nuclei and then measure the amount of nuclear YAP signal, and plotted as a radial distribution. As motioned above, we have now added new supplemental material including highly magnified videos of living D3-D4 neuruloids (Figure 1—video 2, Figure 1—video 3, Figure 1—video 5, Figure 1—video 6, Figure 4—video 2 and Figure 4—video 3) as well as magnification of DAPI/YAP confocal images for WT, WT+TRULI, and HD lines (Figure 1—figure supplement 5).

– Quantifications of fluorescence intensities of the blots would be useful to measure increases in expression.

We would like to thank the reviewer for this suggestion. We now added a Supplemental Figure with quantification of the immunoblot data (Figure 1—figure supplement 4).

– Statistics are needed for the change in expression levels in 2C – is this a significant change? Also what are the Y axes?

Following the reviewer’s advice, we have now added the label for the Y axes as “cell counts", and performed two-way ANOVA and non-parametric multiple *t*-tests on two experiments, showing *p* < 0.000001 for both settings. The statistics have been added to the Figure Legend.

– In 2D they describe nuclear YAP exclusion from the nucleus but it is difficult to discern where exactly the signal is coming from without a DAPI channel or zoomed images of nuclear exclusion.

In agreement with the reviewer, a Supplementary Figure with higher magnification that increases the resolution in different channels has now been added (Figure 2—figure supplement 1).

It would be also interesting to describe the extent of nuclear exclusion and if that correlates with KRT18 expression. Does the more nuclear YAP dictate how much KRT18 expression there is?

Throughout this study KRT18 has been used as a cell-type specific marker to identify early progenitors of the epidermal lineage, in terms of YAP activity and expression. Our study does not address direct regulation of KRT18 by YAP, which is outside of the scope of this study.

– Similar quantifications of cytoplasmic/nuclear expression is needed for Figure 4.

As in Figure 1E, Figure 4B shows the fixed and labeled confocal images of D4 neuruloids (YAP/DAPI), in which the DAPI signal was used to perform nuclear segmentation and then to quantify the amount of nuclear YAP signal in WT and HD specimens.

– The authors describe that verteporfin results in loss of cell adhesion in both WT and HD samples but do not show images describing this phenomenon. As a control it would be useful to show that adhesions are maintained over the course of neuroloid development with and without Verteporfin treatment.

We apologize for the misunderstanding. Following verteporfin treatment during the first three days of the neuronal induction, the cells die and detach from the dish. This is not entirely surprising given that it has been shown that YAP activity is required for early ectodermal differentiation. We have now stated this point more clearly in the manuscript.

Perhaps most importantly, the authors demonstrate that the YAP signature is highly enriched in the NE cluster in HD neuroloids, however, their staining and previous figures show little to no YAP localisation within the PAX6+ NE.

The reviewer is correct, we were able to measure YAP activity by examining the expression of its target genes by scRNA-seq, a methodology that is far more sensitive than imaging. We conclude that the hyperactivation of YAP occurs highly in the NNE at the periphery and is very subtle in the NE at the center. However, because the level of YAP activity in NE is very low, a subtle increase results in a multifold change.

Significant extra analysis of the downstream ramifications of aberrantly increased YAP signalling in the NE is necessary.

While this might be an interesting analysis, it falls outside the scope of our study. Nevertheless, we have now included a robust analysis on the cell cycle effects of YAP activation, and cover potential implications in the Discussion.

The authors claim that increased YAP activity leads to increased NNE and epidermal cell commitment but this is not reflected in the scRNAseq data (Figure 5). It would be valuable to include more robust identification of lineages that includes multiple markers for each lineage (a heat map will illustrate this nicely).

We would like to thank the reviewer for this suggestion, which we have followed. A heatmaps of the top (100 and 20) genes that are differentially expressed most significantly across the three lineages has now been added (Figure 5—figure supplement 4).

– While the authors use TRULI and Verteporfin to study the effects of YAP on neural induction, it would be worthwhile to visualize via fix-and-stains other members of the HIPPO pathway (and especially members downstream of YAP) to show if there are similar effects of nuclear exclusion in WT and HD neuruloids.

As we show in supplemental Figure 5—figure supplement 3, the canonical components of the Hippo pathway upstream of YAP are expressed. Moreover, expression of LATS proteins in the cells where TRULY induces YAP activation suggests that at least some of the canonical Hippo signaling component are at play. Published antibodies for upstream components do not work in our hands, but even so, the non-YAP components of the Hippo pathway do not shuttle into and out of the nucleus, for their activity occurs in the cytoplasm. It is when the Hippo phosphorylation cascade is *inactive*, and YAP remains unphosphorylated, that YAP alone *relocates from the cytoplasm to the nucleus* at an increased rate.

Reviewer #3 (Recommendations for the authors):The authors report exclusion of Yap from nuclei at D3, when neural induction occurs. Is this indicative of initiation of Hippo signalling? Are the upstream stimuli that may lead to Hippo activation here known? Perhaps the authors could speculate on this a bit more in the discussion?

Dissecting the exact upstream regulatory mechanisms of Hippo signaling in this context is very challenging, for the pathway functions as a hub for a plethora of inputs. However, our interpretation is that HD mutation results in abnormal epithelial polarity, which in turn might dysregulate the Hippo pathway. Following the reviewer’s advice, we have now extended this part of the Discussion to address this more fully.

Also, YAP is again upregulated upon BMP stimulation at D4, but is BMP directly activating Hippo signalling here?

We would like to thank the reviewer for raising this issue. The upregulation of YAP at the edge of the colony is induced by BMP4 stimulation. We have added a Supplemental Figure showing that the increase of nuclear YAP at D4 of the differentiation protocol is dependent on BMP4 stimulation (Figure 1—figure supplement 3A). Our video (Figure 1—video 1) shows that the upregulation of YAP at the edge of the colony occurs 18–20 hrs after BMP4 stimulation, suggesting that YAP expression might be directly regulated by SMAD1. To corroborate this hypothesis, we looked into a published dataset of SMAD1 ChIPseq performed in hESC (Tsancov et al., 2015), which shows that SMAD1 binds to the regulatory region of YAP and therefore could directly regulate YAP expression (Figure 1—figure supplement 3B).

Although of considerable interest, the connection between human neurulation and Huntington's Disease could be improved a bit. I wonder if the authors could expand on their discussion of the similarities between TRULI treatment and HD neuruloids. This is a particularly intriguing finding, as the authors state, since HD symptoms manifest only in adulthood. Are embryos carrying the htt mutation less likely to complete development and is there an effect in fertility rates? It would also be interesting if the authors could comment on juvenile HD, which has a much earlier onset of symptoms.

Following the reviewer’s advice, we have now expanded our Discussion to consider more widely the potential developmental implication of HD mutation, including comments on the juvenile forms of HD.

Specific comments:1. Overall, the authors have not presented key experimental details such as the number of neuruloids assessed for each experiment. This information will need to be added to improve confidence in the conclusions.

We would like to thank the reviewer for pointing this out. We apologize for this omission and have now included the number of colonies analyzed in each experiment in the Figure Legends.

2. Perhaps the presentation of Video 1 could be improved to better display the nuclear-cytoplasmic flux of YAP. An enlarged region of representative cells undergoing said flux would be a welcome addition here as would slowing down the video.

As stated above, we apologize for the confusion created by our use of the term “flux”, which we agree can be misleading. We show that following BMP4 stimulation, YAP expression is upregulated at the edge of the colonies and sporadically localizes into the nucleus. Upon TRULI treatment or HD mutation, cells at the edge fail to retain YAP in the cytoplasm, resulting in increased nuclear YAP localization. To clarify this point, we have rephrased the term “flux” and included, as suggested, magnified and slowed videos of YAP-GFP/H2B-mCherry D3-D4 neuruloids (Figure 1—video 2, Figure 1—video 3, Figure 1—video 5, Figure 1—video 6, Figure 4—video 2 and Figure 4—video 3).

3. The data presented in supplementary figure 2 is meant to show the effect of cell density and TRULI treatment on YAP localisation. However, this figure is difficult to interpret as all the panes appear to have similar YAP expression and localisation. Has this data been quantified? An indication of the percentages of cells with nuclear or cytoplasmic YAP would be useful here as would be zoomed images of representative cells.

We would like to thank the reviewer for pointing this out. We have now performed quantitative analysis of these samples, where we calculated the ratios of YAP nuclear/cytoplasmic signals in live cell imaging. The ratios were determined by segmenting nuclei based on the H2B-mCherry channel and using an inverted nuclear mask to obtain the average cytoplasmic signal (Figure 1—figure supplement 2).

4. Figure 2D is difficult to interpret as presented here. For example, it is not clear if any Pax6 positive cells are also expressing YAP. It would be useful to split the channels here.

We would like to thank the reviewer for this suggestion. We have now added a Supplementary Figure with details of D4 to D6 neuruloids in split channels (Figure 2—figure supplement 1).

5. Although it is clear that YAP is present in KRT18 nuclei (Figure 2E), there are a number of other cells in the region that are KRT18 negative but still present nuclear YAP. Are these cells also destined to acquire epidermal fate?

The reviewer is correct. Figure 2E reports representative images of neuruloids at D5 when epidermal fate begins to emerge with few cells to expressing KRT18. We believe that all cells that display nuclear YAP at this stage will eventually become epidermis and express KRT18.

6. On page 6 and in following sections, the authors state that administration of TRULI leads to enhanced YAP activity. However, the data only really demonstrates nuclear localisation rather than activity. Granted, the later scRNA seq experiments then confirm this finding. Perhaps this statement could be toned down here.

We thank the reviewer for pointing this out. In the narrative of the manuscript, we have striven to discriminate localization of YAP from its demonstrated transcriptional activity. However, we believe that nuclear localization of YAP is a good readout of its activity, and is broadly used in the field as such. Additionally, and as the reviewer notes, we do have scRNA-seq results that show enriched activity of YAP target genes in the epidermal lineage; we also found strong YAP nuclear localization through immunohistochemistry and live imaging.

7. The control images shown in figure 3 appear to also possess a lumen, albeit this is smaller than the lumen of the TRULI treated neuruloids. Is this indicative of slower neurulation compared to controls? Should neurulation be complete at the stages shown? I assume this is the case as the images shown in figure 2D show fully closed lumens.

The phenotype observed in neuruloids is due to both on the relative abundance of discrete lineages (increased epidermis and reduced NC), and on the structural level (failure in closing of the central NE domain). To clarify the point raised by the reviewer we now added a diagram (Figure 6E) to represent the structural aspect of the HD/YAP activation phenotype as well as a paragraph in the Discussion that addresses this point.

8. The authors report that administration of verteporfin in the first three days of culture leads to loss of cell adhesion, but there is not data shown to support this.

We apologize for the confusion. Verteporfin treatment during the first three days of neuruloid differentiation leads to cells death, leading to their detachment from the dish. To better clarify this point we have rephrased that line to be more explicit.

9. The methods section should also include a section on immunoblotting.

We have now included a methods section for immunoblotting and the appropriate reference (Vrbský, et al., 2021)

We would like to thank all three reviewers for taking the time to evaluate our manuscripts and for their suggestions and comments, which we believe will strongly improve the quality of our study.